# Human Health Risks due to Exposure to Water Pollution: A Review

Preethi Babuji [1], Subramani Thirumalaisamy [1,*], Karunanidhi Duraisamy [2] and Gopinathan Periyasamy [3]

1 Department of Geology, College of Engineering Guindy (CEG), Anna University, Chennai 600025, India; bpreethi411@gmail.com
2 Department of Civil Engineering, Hindusthan College of Engineering and Technology (Autonomous), Coimbatore 641032, India; karunasamygis@gmail.com
3 CSIR-Central Institute of Mining and Fuel Research (Ministry of Science and Technology, Government of India), Dhanbad 828108, India; srigopi555@gmail.com
* Correspondence: geosubramani@gmail.com; Tel.: +91-967-737-7554

**Abstract:** Water resources are crucial in developing any area as they serve as a major source of potable, agricultural, and industrial water. Water contamination, caused by natural and anthropogenic activities, poses a significant threat to public health globally. This review synthesizes data from various studies published in national and international journals, as well as reports from governmental and non-governmental organizations. Our primary objective is to understand and review previous research on water pollution, contamination types, and the effects of water contamination on public health. Water pollution studies generally involve a scientific understanding of the biological, chemical, and physical processes that control the movement of contaminants in the underground environment. The nature and severity of health consequences vary based on several factors, including the chemical composition, duration of exposure, and concentration of pollutants. This work highlights the human health risks associated with current research topics such as anthropogenic, geogenic, microplastics, pharmaceuticals, and heavy metals. A section on remedial measures and mitigation strategies is included to emphasize sustainable approaches to water conservation, replenishment, and sustainability. However, there is a lack of comprehensive knowledge regarding the distribution, toxic effects, and human health risks associated with different sources of contamination. This review thus establishes links between multiple sources of pollution, their toxicity to human health, and approaches to health risk assessment.

**Keywords:** water pollution; contamination types; health risk assessment; toxicity; mitigation strategies





## 1. Introduction

Concerns regarding water pollution are widespread as it damages people's health and well-being. To support life, improving public health, halting the development of waterborne illnesses, and access to clean, safe water are crucial. However, water quality may be harmed by several contaminants, rendering it unfit for daily use and consumption.

Human and natural activities both have the potential to contaminate water. Natural sources can be found in microbial activity, geological structures, and naturally existing pollutants found in water supplies. On the other side, anthropogenic causes are caused by human activities such as industrial operations, agricultural practices, inappropriate waste disposal, and insufficient sewage systems. These practices contaminate and pollute water supplies, jeopardizing their safety and purity.

Water contaminants might include microbiological diseases, chemical pollutants, heavy metals, pesticides, medicines, and new contaminants. Water sources can be contaminated by chemical contaminants, including home chemicals, agricultural runoff, and industrial waste, which are harmful to human health. Their effects on human health are

continuously being researched, and new pesticides, medications, and developing pollutants pose further difficulties.

Health risk refers to the probability of encountering a hazardous substance that can potentially cause illness in humans. The assessment of health risk involves the multiplication of two factors: hazard and exposure [1,2]. The process of determining health risk comprises four main steps: hazard identification; exposure assessment; concentration determination; and using a mathematical model to evaluate the human health risk (HHR) based on exposure and dose–response evaluations [3]. To accurately predict the adverse effects on human health resulting from various situations, it is essential to have specific information for each pollutant, including a baseline incidence of morbidity or death, as well as concentration–response curves derived from studies on the health effects of the specific pollutant [4]. This information is vital for accurately predicting trends in negative health effects caused by different scenarios.

The issue of human health in different locations globally is closely associated with environmental and groundwater pollution [5–7]. Additionally, improper handling and the persistence of plastic trash result in the buildup in the environment of microplastics, the transmission of pollutants, and the leaching of hazardous additives [8]. Due to their proximity to the chemicals, heavy metals, drugs, pesticides, and other persistent organic contaminants previously stored in them, microplastics are frequently referred to as a combination of harmful agents [9–11].

The deterioration of water quality is a growing concern due to significant environmental changes and increasing human activity. Water quality is affected by various factors, such as natural processes, human activities, and climate change. The impact of these factors on water quality can be seen in the form of increased levels of pollutants and contaminants in water bodies. The presence of these pollutants and contaminants can have serious implications for human health and the environment. Therefore, it is important to monitor water quality and take appropriate measures to prevent further deterioration [12–14]. Anthropogenic factors such as the improper disposal of pharmaceuticals, metabolic excretion, industrial use, and municipal sewage have contributed to this decline [15]. Numerous research has been undertaken globally to investigate groundwater quality and related health hazards since drinking contaminated water can have serious negative effects on human health [16–20]. These studies have linked contaminants found in groundwater to various health issues, including but not limited to obesity, diabetes, cancer, endocrine disruption, cardiovascular, developmental problems, and reproductive issues [21–24]. The health implications of water contamination are numerous and varied. Exposure to contaminated water can result in acute or chronic health effects, depending on the specific contaminants and the duration and level of exposure. Waterborne diseases can cause gastrointestinal disorders, dehydration, and even life-threatening conditions, particularly in vulnerable populations such as children, the elderly, and individuals with weakened immune systems. Long-term exposure to certain contaminants can lead to organ damage, developmental issues, reproductive problems, and increased cancer risk.

This review, "**Human health risks due to exposure to water pollution: A Review**", is a compilation of several case studies on human health risk assessment and its associated dangers in connection to various types of pollutants and sources in various regions of the world. In this paper, multidisciplinary tools are used to evaluate the risk to humans on numerous sources of contamination. Investigating the causes, kinds, and effects of water pollution on health is the goal of this study. This study intends to further our awareness of this crucial topic and contribute to creating plans and actions aimed at reducing the dangers of water pollution and preserving human health. It does this by analyzing the body of literature and research findings.

## 2. Data Collection

This review paper studied different scientific journals and articles related to water contamination and its effects on human health. The studies about the sources of water

contamination were screened into different contamination aspects like Anthropogenic, Geogenic, Pharmaceuticals, Microplastics, and Heavy Metals. The papers screened as such were sourced globally with a primary focus on human health risk aspects. Based on the sampling methodology, source, contamination extraction procedures, detection techniques, elemental concentration, age and gender categorization, recommendation, and remediation technologies for sustainable management of water resources, an effort to consolidate the information has been attempted. A review of several industrial zones, mines, agricultural waste, pharmaceutical waste, sewage, and lithology was undertaken to study water contamination-induced health risks. Figure 1 depicts the different types of contamination for health risk assessment from different countries: (1) Anthropogenic; (2) Microplastic; (3) Geogenic; (4) Pharmaceuticals; (5) Heavy metals.

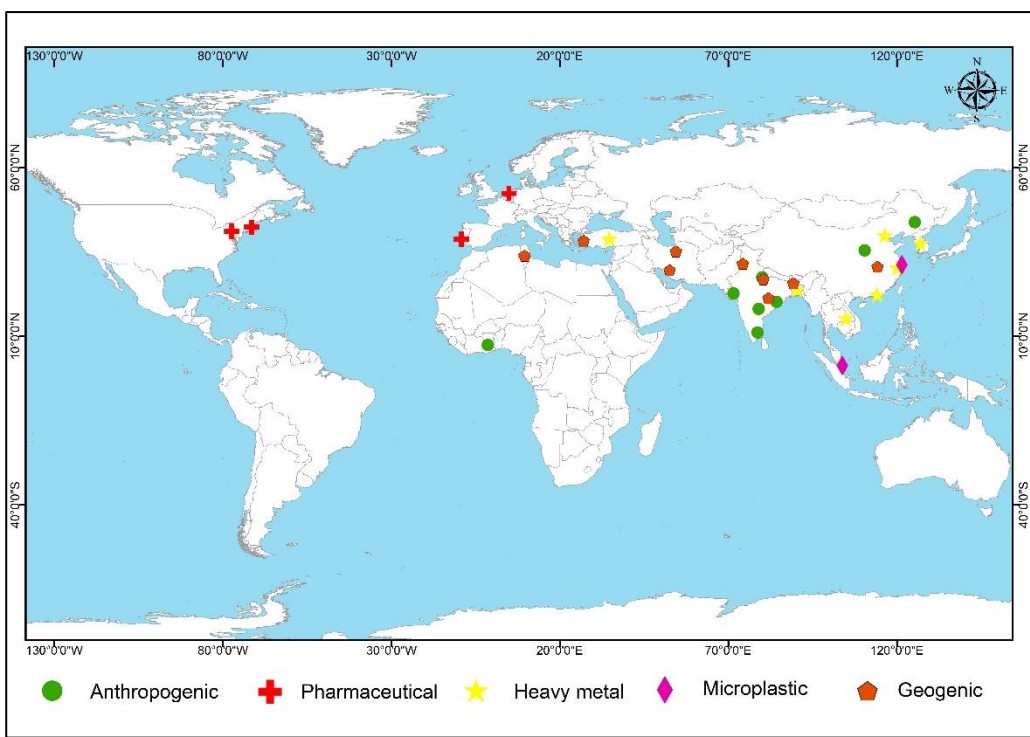

**Figure 1.** Different sources of contamination from different parts of the world.

## 2.1. Anthropogenic Factors

The relationship between human activities and groundwater quality and the associated health risks was examined in a collection of nine publications. One of these studies [25] focused on the impact of heavy mineralization and industrialization in the Subarnarekha River basin, India, on groundwater contamination by metals. This region is a significant mining and industrial area in India, where minerals such as copper, iron ore, and uranium are extracted, and steel, aluminum, cement, and power are produced, among other related activities.

According to the study, there were significant regional variations in the number of dissolved metals and metalloids present in the groundwater of the Subarnarekha River Basin. The World Health Organisation (WHO), India, and the United States Environmental Protection Agency (USEPA) have defined drinking water standards and at several of these locations, arsenic, manganese, copper, iron, and selenium levels surpassed these standards. According to the study, manganese, cobalt, and arsenic were identified as the main causes of long-term non-carcinogenic hazards.

Additionally, the researchers estimated the Hazard Quotient (HQ) and Hazard Index (HI) for each area. They found that some locations had HQ values for adults for manganese, arsenic, and cobalt and children for arsenic, manganese, cobalt, selenium,

vanadium, zinc, and iron that were greater than unity. The tolerable cancer risk threshold of $1 \times 10^{-4}$ was met by arsenic for both adults and children. The data revealed that greater metal concentrations were close to industrial and mining facilities, indicating that human activities have substantially impacted their concentrations.

Adimalla et al. [26] investigated the groundwater quality and the health dangers posed by $F^-$ and $NO^{3-}$ poisoning in the Nirmal Province of South India, where the major source of drinking water is groundwater. The main causes of the elevated nitrate concentration in the region were determined to be overuse of fertilizers and manures, septic tank leaks, and high amounts of organic waste. The non-carcinogenic health hazards were $2.95 \times 10^{-1}$ to $4.07 \times 10^0$ for males, $3.49 \times 10^{-1}$ to $4.80 \times 10^0$ for women, and $3.99 \times 10^{-1}$ to $5.50 \times 10^0$ for children, respectively. The allowed total health index (THI = 1) for adults, adolescents, and children was also surpassed in a significant number of groundwater samples. Since children are more vulnerable to health hazards than men and women in the research region, this was shown by the health risk assessment.

In a different research, D. Karunanidhi et al. [27] assessed the groundwater's appropriateness for a semi-urban region of south India's food production and sustainable drinking water supply. According to the study's risk assessment, the groundwater found in the semi-urban area of southern India is very saline and just slightly alkaline. TDS, Cl, and Na showed a positive association, showing their substantial influence on groundwater pollution, which the study linked to human activities. A significant number of samples above the threshold limit of 1 for babies, adolescents, teens, and adults, according to the Total Hazard Index (THI) calculated to assess the health hazards, were related to fluoride and nitrate in groundwater. The study suggested using groundwater effectively to accomplish SDGs 3, 6, and 11 for sustainable development.

A case study [28] was undertaken on nitrate pollution in a remote area of northeast China's groundwater. The analysis showed that the study area's drinking water contained high concentrations of nitrates that presented a threat to people's health. A large number of samples included nitrate, and in a third of the samples, abnormal concentrations were discovered. According to the study, locations near agricultural areas and sewage irrigation canals were more likely to have nitrate contamination of the groundwater than urban areas, which saw a reduced risk. The study also showed that children were more likely than adults to have negative health impacts from nitrate exposure. According to research, groundwater contaminated with Cr had an anthropogenic origin [29]. According to the study, Cr levels were somewhat extremely low in peri-urban regions yet extremely high in urban areas, indicating that industrial sources are the cause of the pollution. During pre-monsoon, lead (2.4) and cadmium (2.1) had Hazard Quotient (HQ) levels for children above the safe limit (HQ = 1), but post-monsoon values only applied to lead (HQ = 1.23). The findings of this study showcased that children in the area may experience health concerns from prolonged exposure to polluted groundwater, which is also unfit for human consumption.

Using pesticides applied in Ejura, Ghana, a subsequent study [30] evaluated the health risks associated with eating maize and cowpea. In all, 37 pesticides, including pyrethroids, organochlorines, and organophosphorus insecticides, were found in samples of cowpea and maize taken from farms near Ejura. According to the health risk calculation, some of the pesticides discovered in maize and cowpea surpassed the Acceptable Daily Intake, raising the possibility of chronic toxicity for those who consume these foods.

The dangers to human health and the environment posed by radionuclides in South Indian limestone mining zones were estimated in the research [31]. The study discovered a greater value of excess lifetime cancer risk (ELCR) recorded at the sample sites and found that the estimated average absorbed dose rate of the soil samples was marginally higher than the world's mean. This implies that mining sites should be recovered following mineral extraction and human traffic, and planting crops should be done carefully to prevent further lengthy exposure.

In Hancheng City, in the Guanzhong Plain in China, study [32] investigated the seasonal variations in the water quality for residential use. According to an analysis of many

factors, over 80% of the water samples in the research were of excellent quality and fit for consumption and other household uses. Nevertheless, the study did evaluate several pollutants' potential carcinogenic and non-carcinogenic health hazards. It concluded that for both adults and children, the non-carcinogenic health risks are higher in the dry season than in the rainy season. Compared to adults, children are almost twice as likely to develop cancer.

In rural Yantai, China, research [33] examined an analysis of the health risks posed by groundwater nitrate pollution. In the Jiaodong region, which includes Yantai, Weihai, and other cities, the study discovered a high nitrate level. This nitrate level may be due to the extensive use of fertilizer to produce fruit trees. According to the study, both adults and juveniles had Hazard Quotients (HQ) higher than 1, and adult females were shown to be more susceptible to groundwater nitrate pollution than adult men. Even without taking the dermal contact route into account, children have the highest exposure risk.

### 2.2. Microplastics

To gain insights into the impact of microplastics on human health, two research papers were examined. Study [34] investigated the prevalence of microplastics in various environments, including terrestrial and marine ecosystems, and analyzed different organisms to ascertain the Global Average Rate of Microplastic Ingestion (GARMI). The study's findings showed that a variety of factors, including particle size, shape, and type of polymer, as well as the distribution of particle sizes, all affected how many microplastics were ingested. The findings indicated that groundwater is the main source of the 11,845 to 193,200 microplastics that individuals may swallow annually on a worldwide scale. The study emphasized the possible hazards to human health from the continued use of microplastics. The investigation also showed that, depending on several variables, including individual age, size, demography, traditions, geography, environmental development, and lifestyle choices, the average weekly intake of microplastics ranged from 0.1 to 0.5 g.

Another study [35] focused on a specific geographic location, the Changjiang Estuary and the adjacent East China Sea, and evaluated the concentration and chemical composition of microplastics in collected samples. The study assessed the MPs' pollution load index and determined that the study region was moderately contaminated. Fisheries were selected as high-risk sites. It was discovered that the MP's degree of risk contamination was related to both human activity and hydrodynamic dynamics. The study emphasized the need for improved legislation and regulations to reduce microplastic pollution. Table 1 describes the formulae used by authors to estimate microplastics in water.

**Table 1.** Formulae used by different authors for microplastic contamination.

| S No. | Author | Formula | Abbreviation |
|---|---|---|---|
| 1 | [24] | GARMI = (ANMP) $\times$ (AMIMP) | GARMI—the global average rate of microplastics ingested<br>AMIMP—average mass of an individual microplastic particle<br>ANMP—the average number of microplastic particles |
| 2 | [25] | (a)　$H = P_n \times S_n$<br>(b)　$CF_i = C_i / C_{oi}$<br>(c)　$PLI = \sqrt{CF_i}$ | H—polymer risk index caused by MP<br>$P_n$—percent of MP polymer types<br>$S_n$—score for the polymer compound<br>$CF_i$—MP concentration factors<br>$C_i$—MP concentration at each station<br>$C_{oi}$—minimal MP concentration<br>PLI—pollution Load Index |

### 2.3. Geogenic Factors

Eleven papers addressed the impact of geology and lithology on water quality and the accompanying health implications. In one study [36], a health risk assessment was conducted on fluoride in Central Europe, revealing that the regions with the highest risk of

fluoride exposure were located in Ukraine and Moldova. The study found that geology can be used to create a basic risk assessment scheme. However, caution is necessary as geological maps are only two-dimensional representations and may not provide accurate indications of fluoride risk. Additionally, deeper waters often contain more fluoride than shallow waters. Dental and skeletal examinations in the study showed that water with a fluoride range from 1.5 mg/L to 5 mg/L caused dental fluorosis but did not significantly impact bone tissue.

In another study [25], high levels of metalloids were discovered in the groundwater of India's Subarnarekha River Basin, suggesting an unusual geochemical enrichment originating from geological and anthropogenic sources. The study identified Mn as the most significant pollutant, as exposure to high levels of this metal over several years can lead to nervous system toxicity similar to Parkinsonism. The main causes of chronic hazards were manganese, cobalt, and arsenic, and depending on the average geometrical concentration of metals and metalloids, the Hazard Index (HI) for individuals was higher than unity. This investigation concluded that metals in drinking water constituted a risk for oral absorption. The amount of arsenic in groundwater and the ensuing dangers to human health were calculated through chronic daily intake, Hazard Quotient (HQ), Hazard Index (HI), and carcinogenic risk (CR) for both oral and superficial exposure to arsenic, according to research by [37]. The study indicated that arsenic enrichment in groundwater in the studied areas is mainly because of geogenic emissions, such as weathering of parent minerals like arsenopyrite and realgar. Study [38] indicated that when arsenic is consumed orally and via the skin, there is a significant danger that it may cause cancer in people. The values for both the cancer index (CI) and CR were found to be greater than the United States Environmental Protection Agency limit ($10^{-6}$). The study concluded that people's exposure to arsenic through drinking water has both cancer-causing and non-cancerous consequences on their health, and it urges immediate management and corrective measures to safeguard them against arsenic.

In another study, ref. [39] reported geogenic contamination of copper and manganese in groundwater by applying PCA in the research field. Based on their significant negative loadings, the study hypothesized that each of these two elements could have a natural origin. Furthermore, the samples' high hardness level results from the inclusion of carbonate rocks. The study identified possible health hazards for kids from prolonged exposure to polluted groundwater. According to this study, children are more in danger than adults from consuming this water. Before the monsoon, the Hazard Quotient (HQ) for adults and kids was found to be as follows: Pb > Cd > Ni > Zn > Cu > Mn > Cr and Pb > Cd > Ni > Zn > Mn > Cu > Cr, respectively. After the monsoon season, the sequence of elements was Pb > Cd > Zn > Ni > Cu > Mn > Cr for adults as well as children.

Most likely, the fluoride in the area comes from local shale, according to geochemical provenance techniques used by [39] to identify major and rare earth elements in soils. According to the study, chemical weathering and evaporation both affect the chemistry of groundwater. Children were found to have a greater average computed fluoride exposure from drinking water (0.082 mg/kg/day) than adults (0.047 mg/kg/day) or adolescents (0.046 mg/kg/day). The ideal daily fluoride intake from all dietary sources is established at 0.05 mg/kg/day. Due to excessive fluoride exposure from drinking water alone, residents in the research region are at risk for dental fluorosis. Surface morphological changes in enamels with fluorosis can result from the presence of fluoride. Fluoride occurrences might thus result from aquifer elements that are already present in the groundwater system under study.

Ref. [40] evaluated the potential global danger to human health caused by PTE contamination levels in soils near uranium mines. The study discovered that the average level of PTEs in the tested soils exceeded their corresponding world average levels, with cadmium and uranium being the main influences on the ecological settings of uranium mining sites. According to the health risk assessment, the major way that PTEs in soils are exposed is by oral intake, and uranium and arsenic may provide particularly substantial non-carcinogenic

hazards to nearby children. These findings emphasize the need to minimize health hazards for locals, especially kids, and limit soil contamination in uranium mine-associated soils.

The detrimental effects of consuming drinking water from the province of Izmir, Turkey, while exposed to trace metals were evaluated in [41]. The study discovered that arsenic's cancerous risks were >10 for 46% of the population and >10 for 90%. In comparison, arsenic's non-carcinogenic hazards were greater than the threshold of concern for 19% of the population.

Another study conducted in Iran [42] investigated the number of trace elements in raisin samples, including lead, arsenic, cadmium, nickel, copper, zinc, and iron, as well as residual sulfur dioxide. Except for two samples, the study determined that Iranian raisins pose no significant health danger to consumers. It calculated the sulfur dioxide residue exposure's negative effects on human health. The study suggested more in-depth research to comprehend the trace components' impact fully.

Uranium contamination of the groundwater in the Bemetara area of Chhattisgarh state was assessed in an investigation by [43] in India. The investigation discovered that various metrics, including TDS, TH, Ca, Mg, and Cl levels, were greater than the permitted limit established by WHO 2011/BIS 2012. However, cancer risks associated with water intake in the research location were substantially lower than the acceptable level. The report suggested installing a water purification system to filter the water.

According to the age of the customers, study [44] evaluated the risk associated with fluoride exposure from drinking water in Tunisia. According to the study, over 75% of Tunisians may be in danger of dental decay, 25% may be at risk for dental fluorosis, and 20% may be at risk for skeletal fluorosis.

The last research investigated the spatial and temporal distribution of fluoride in drinking water, primarily near the shore, in Bangladesh [45]. According to the study, in both seasons, children and infants who drank water with a high fluoride content had non-carcinogenic hazards that, on average, were higher than the threshold value of 1 (HQ > 1). The study also showed that, in addition to excessive exposure, fluoride shortage may pose a serious issue in this area due to the extremely low fluoride content in drinking water.

### 2.4. Pharmaceuticals

Four papers evaluated the health hazards associated with pharmaceuticals in the context of environmental exposure. The possible effects of 44 active pharmacological compounds in fish intake or drinking water on human health were examined in one investigation by [46]. The point of departure (POD) for the ADI was determined using the acceptable daily intake (ADI), which was derived from the lowest daily therapeutic dose or the lowest effect level (LOEL) or no-effect level (NOEL) from pre-clinical toxicology investigations. Predicted no effects concentrations (PNEC) were calculated for each medication active in children and adults using the ADIs and common assumptions regarding fish exposure and water consumption. All the compounds were found to have ratios less than one, ranging from $7 \times 10^{-2}$ to $6 \times 10^{-11}$. Based on the information at hand, a conclusion is that fish and water consumption pose no significant danger to the health of people from exposure to the environment.

Using the following formula, ref. [47] calculated lifelong exposures to pharmaceuticals through drinking water: concentration of pharmaceutical (ng/L) drinking water consumption (L/d) 65.25 (d/y) 70 (y)/1,000,000. This study focuses on the presence of medicines in Dutch drinking water. It was discovered that drinking water derived from the Rhine and Meuse rivers had higher quantities of medicines than water obtained from polder water. According to research, groundwater is less likely than surface water to become contaminated with residues of human-made micropollutants. Only a few milligrams, or less than ten percent of the total dose given to a patient in a single day, were predicted to account for a patient's lifetime exposure to drugs by drinking water. It was determined that there was very little danger of negative health impacts.

Based on environmental monitoring data from the United States, risk assessments for human health were carried out for 26 active pharmaceutical components (APIs) and/or their metabolites in another research [48]. In this study, the potential for direct exposure to trace amounts of APIs in US surface waters to have an impact on human health was assessed. Due to their larger ingestion of water and fish, children's PNECs were shown to be less than those of adults. The investigation concluded that the mere existence of trace amounts of each of these APIs in drinking water and surface water poses no major danger to human health.

To improve the accuracy of the risk assessment, ref. [49] conducted a surveillance investigation on 31 pharmaceuticals along Lisbon's drinking water supply and evaluated the hazard to human health by employing risk quotients (RQs) based on various life stages. Although small amounts of medicines were found in water for consumption, their threat to human health was determined to be negligible based on recent toxicity data. Infants 0 to 3 months old were discovered to have a greater risk quotient. Table 2 depicts the formulae used by authors for estimating the pharmaceutical contamination in water.

**Table 2.** Formulae used by different authors for pharmaceutical contamination.

| S No. | Author | Formula | | Abbreviation |
|---|---|---|---|---|
| 1 | [39] | (a) (b) (c) | $\text{PNEC}_{dw} = \frac{1000 \times ADI \times BW \times AT}{Ing\ Rdw \times EF \times ED}$ <br> $\text{PNEC}_{fish} = \frac{1000 \times ADI \times BW \times AT}{BCF \times Ing\ \ Rf \times EF \times ED}$ <br> $\text{PNEC}_{HH} = \frac{1000 \times ADI \times BW \times AT}{Ing\ Rdw \times BCF \times Ing\ Rp\ EF \times ED}$ | PNEC—predicted no effects concentrations (ng/L) <br> *ADI*—acceptable daily intake (µg/kg/day) <br> *BW*—child or adult body weight (kg/person) <br> *AT*—averaging time (days) <br> *Ing Rdw*—child or adult drinking water ingestion rate (L/person/day) <br> *Ing Rf*—child- or adult-fish consumption rate (kg/person/day) <br> *BCF*—bioconcentration factor for fish (L/kg) <br> *EF*—exposure frequency (days/year) <br> *ED*—exposure duration (years) |
| 2 | [41] | (a) | $ADI = \frac{1000 \times POD}{UF1 \times UF2 \times UF3 \times UF4 \times UF5}$ | *ADI*—acceptable daily intake (µg/kg/day) <br> *POD*—point of departure in (mg/kg/day) <br> *UF*—unitless uncertainty or modifying factors |
| 3 | [42] | (a) (b) | $RQ = \frac{Cs}{DWEL}$ <br> $DWEL = \frac{ADI \times BW \times HQ}{DWI \times AB \times FOE}$ | RQ—risk quotients <br> $C_S$—concentration of the pharmaceutical compound <br> DWEL—drinking water equivalent level <br> *ADI*—acceptable daily intake (µg/kg/day) <br> *BW*—child or adult body weight (kg/person) <br> *HQ*—Hazard Quotient <br> *DWI*—drinking water intake (L/day) <br> *AB*—gastrointestinal absorption rate <br> *FOE*—frequency of exposure |

*2.5. Heavy Metals*

According to research on the assessment of health risks associated with heavy metal and metalloid pollution near the defunct Songcheon Gold-Silver mine in Korea, which looked at a total of nine studies on the subject [50], the Korean Ministry of Environment's permitted level of water for drinking standard was found to be exceeded by the concentration of As and other heavy metals in stream water and groundwater in 2003. The levels of Cu and Pb, however, were acceptable. The mine site's Hazard Quotient (HQ) rating was 16, and As's Hazard Index (HI) score was 15. The carcinogenic risk was $2.7 \times 10^{-3}$, which indicates that there is a likely chance that 3 cancer patients per 1000 individuals may develop. This number was higher than the typical range of $10^{-4}$ to $10^{-6}$.

The occurrence of heavy metals in soil–plant systems is being investigated using several media described in another research [28]. The intake dosages of heavy metals were calculated using three sources: soil; plant; and groundwater. According to the health risk assessment's findings, nutrition accounted for the majority of exposure routes, and heavy metals found in soil samples might potentially affect humans through the food chain. The overall non-cancer and cancer-related risk results showed that the examined arable fields close to waste mining and industrial sites were inappropriate for producing leaf and root crops due to the danger of increased absorption of heavy metals negatively



influencing food safety for the local population. According to the study, non-cancer risks were primarily associated with chromium and lead, while cadmium posed the highest cancer risk. The findings suggested that to mitigate pollution in the studied region, greater attention and measures should be directed toward controlling the levels of cadmium and chromium [51]. An evaluation was conducted to determine the dangers of heavy metal poisoning in groundwater to the human health of the Hua-rural sub-district located in the Ubon Ratchathani province of Thailand. The study found that the concentration levels of the detected heavy metals in each well, as well as the overall average, fell within the acceptable limits of groundwater standards for cadmium, chromium, copper, mercury, nickel, and zinc. However, elevated levels of lead were observed in certain wells. In 58% of the wells assessed, the Hazard Index (HI) values surpassed the acceptable thresholds. The study identified higher HI results for groundwater wells situated within intensively cultivated chili fields, exceeding one. These findings suggested that individuals residing in warmer regions are more vulnerable to the risks associated with groundwater pollution, given their elevated daily water consumption. This could potentially result in a rise in cases of both cancer-causing and non-cancerous health issues among locals exposed to heavy metals present in the groundwater through drinking. A study conducted by [52] in Lahore, Pakistan, assessed the levels of heavy metals present in contaminated vegetables from various irrigation sources. The research indicated that heavy metal concentrations in groundwater were within the acceptable limits set by Indian standards. However, wastewater used for irrigation of food crops, particularly vegetables, contained higher concentrations of copper, nickel, chromium, cadmium, lead, manganese, and cobalt, exceeding the Indian permissible limits. The study's findings revealed that the bioavailable levels of heavy metals in soil irrigated with wastewater were higher in comparison to soil irrigated with groundwater. During their examination of the surface and groundwater resources in an important agricultural and industrial region in Turkey, ref. [53] identified the presence of toxic metals. The study utilized the Heavy Metal Pollution Index (HPI) as a metric for evaluating the potential risks posed by the investigated toxic-trace elements. The results indicated that the groundwater resources were at the greatest risk from chromium, lead, nickel, cadmium, zinc, and copper, with chromium being the most concerning. Previous studies have shown that the usage of insecticides that contain lead can result in significant environmental contamination, and these pesticides may easily infiltrate the soil and crops, ultimately contaminating surface and groundwater through precipitation and incorrect irrigation methods (ATSDR 2012a). The findings of this study emphasize the need for more effective measures to control and prevent heavy metal pollution in regions where agriculture and industry coexist. According to [54], industrial activities are considered the primary anthropogenic source of chromium found both in the surface and groundwater. In the Ergene River Basin, the study evaluated the Hazard Quotient (HQ) and Hazard Index (HI) coefficients for both adults and children separately, taking into account the intestinal and dermal effects of all investigated toxic elements. Additionally, the study calculated the cancer risk (CR) for chromium, lead, and cadmium in terms of gastrointestinal absorption of toxicants for adults and children. These findings underscore the importance of evaluating the possible dangers of heavy metal contamination in surface and groundwater sources and the need for appropriate measures to control and prevent it. The study's findings indicated that chromium posed the highest risk, based on the calculated Hazard Quotient (HQs), Hazard Index (HI), and cancer risk (CR). The Ergene River and Çorlu Stream were identified as the riskiest habitats for the basin, emphasizing the urgent need for effective measures to control and prevent heavy metal pollution in these areas. Chinese mining contamination from heavy metals and health risk evaluation were examined by [55]. The Geoaccumulation Index was used in this work to evaluate the possible hazard levels, together with the approach for evaluating health risks suggested by the US Environmental Protection Agency (USEPA). The priority control mining categories include tungsten, manganese, lead–zinc, and antimony mines, while the priority management heavy metals are Cd, Pb, Cu, Zn, Hg, and Ni. According to this study, the predominant exposure channel for Cd, Cr, Cu, Ni, and

Hg was dermal absorption, whereas the most typical exposure mechanism for Pb and Zn was ingestion. Due to their high levels in the nearby soil or low RfD values, Cd, Ni, and Pb were the metals to which people were most exposed. The study concluded that mining operations might endanger the population, particularly young children, with significant non-carcinogenic hazards. To research the distribution, accumulation, and evaluation of the health risks associated with heavy metals, ref. [56] gathered dust from the streets of areas of Beijing, China, with high traffic density, residences, educational institutions, and tourist destinations. Cd, Pb, Zn, Cu, Cr, and Ni were the heavy metals with the highest levels of contamination, according to a Geoaccumulation Index pollution assessment. The Cd values qualified as being "heavily contaminated". The health risk assessment methodology used to determine human exposure revealed that, except for children, both the non-carcinogenic and cancer-causing risks of certain metals in the dust from streets were generally in the low range. Ref. [57] examined the possible hazards to one's health from transferring metals from soil to produce in Bangladesh. One of the primary routes for (heavy/trace) metals to enter the body is through the consumption of vegetables. The findings revealed that the level of hazardous substances like Co, Cu, Mn, Pb, Se, Ni, V, and Zn in vegetable samples was below the World Average value, while that of Si, Ba, K, Ca, Mg, Fe, Sc, V, Cr, Cu, Zn, Mn, Co, Ni, Se, Sr, Mo, and Cd was higher than the World Average value in soil and lower than the World Average value for Al, Ti, and Pb. The number of hazardous metals (Fe, Cu, Mn, Zn, Co, Cr, V, Ni, Pb, and Cd) that may be consumed from vegetables is not excessive and is far below the allowed level advised by the US EPA, Food & Nutrition Board, and WHO. Calculated Hazard Quotients (HQ) of iron, copper, cobalt, chromium, vanadium, nickel, lead, manganese, zinc, and cadmium indicated that these metals are most hazardous in the following order: Cd > Mn > Zn > Pb > Cu > Fe > Ni > V = Co > Cr. The highest HQ value for Cd was determined to be 2.543, which is over the safe level.

### 3. Remedies and Mitigations

- **Natural Bioremediation:** A passive or inherent type of remediation that uses natural processes to remove contaminants from groundwater. Additionally, this mechanism changes the pollutant. The pollutant is changed through advection, disintegration, co-metabolism, adsorption, diffusion, and dispersion [58]. Natural in situ biological remediation has the advantage of using native microorganisms. It is, therefore, less expensive than developed bioremediation since there is no need for modification, and the microorganisms modify to their natural surroundings until oxygen and levels of nutrients reach their limiting levels [59].

- **Engineered bioremediation:** Utilizes constructed systems that deliver nutrients, attract electrons, and/or other proliferation-stimulating elements. It is a method of remediation that boosts the development and degradative activity of microorganisms. Biostimulation, bioaugmentation, bioventing, biological permeable reactive barrier (PRB), and phytoremediation are some of the procedures used in this method.

  (a) **Bioaugmentation:** A technique that uses immobilized, genetically stable, or free microorganisms to function as a culture, destroying pollutants and forcing them to endure unfavorable conditions. To increase their functional capability, these specialized bacteria may move. The result of this process is heavily reliant on how local microorganism groups respond to the presence of these genetically modified microorganisms (GMM) or nonindigenous species since they are encapsulated in the groundwater plume that must be repaired. This is because the nutrients they contain are crucial to this process [60];

  (b) **Biosimulation:** A technique for accelerating the rate of bioremediation. By changing the groundwater's chemical and physical characteristics, bioremediation is sped up in this procedure. To boost the biological absorption of electron donors, several nutrient sources are added to the polluted groundwater, including biogas, dung, slurry, and other organic material. When it comes to eliminating hydrocarbon petroleum, comparative research found that biostim-

ulation performs better than bioaugmentation, but when they are combined, they produce the best results in the quickest length of time;

(c) **Bioventing:** The process involves oxidative-biological remediation and soil venting to remove light and moderate distillate hydrocarbons from the groundwater's vadose zone [61];

(d) **Bioslurping:** The adaption and use of vacuum-enhanced dewatering methods to rehabilitate hydrocarbon-contaminated locations. To address two different pollutant media, it makes use of components of both biological ventilation and free product recovery;

(e) **Permeable reactive barrier (PRB):** A zone of in situ treatment that passively traps a plume of pollutants, removes or degrades the contaminants, and releases uncontaminated water. Swelling and precipitation, chemical reactions, and biological mechanism-based reactions are the three main techniques for elimination;

(f) **Phytoremediation:** A technique that uses interactions between plants and pollutants to lessen the hazardous effects of toxins in polluted groundwater. Numerous processes, including filtering, accumulation, decomposition, volatilization, and stabilization, are involved in this process [62].

To assist in the prevention of water pollution, many mitigation measures also need to be implemented in addition to remedial procedures. The following are a few water mitigation strategies:

- To increase the amount and quality of groundwater in the study region, the best rainfall recharge solutions for an urban context must be put into practice;
- To stop future contamination, regulate wastewater outflow, and improve mining waste management;
- To lessen groundwater contamination, septic tanks and sewage systems must undergo routine maintenance;
- Reusing residential wastewater will benefit from the building of biological treatment facilities;
- To prevent the production of leachate, reusable solid wastes ought to be treated separately, and municipal solid waste disposal yards should be constructed with suitable lining,
- Rigid management of environmental effect regulations and improvised government regulations.

## 4. Conclusions

Water contamination has become a serious issue in the current century. With the advancements in technology, the concern regarding water pollution remains underrepresented in many parts of the world. This could be due to a lack of knowledge about the sources of water contamination and its harmful effects on health, particularly by. This review paper supports awareness based on such aspects. From the review studies made on water contamination caused by anthropogenic activities, some of the major contamination sources are fertilizers, pesticides, and industrial and mining activity. Studies have reported that nitrate, harmful chemicals, and other radioactive elements reduce both surface and groundwater quality. Children and pregnant women are subject to higher health risks from water consumption from such contaminated sources. Studies show that sea/freshwater food products could be one of the main sources of microplastic intake. Microplastics dispersed in marine and freshwater bodies are caused by littering and improper waste disposal strategies implemented by local authorities. Sewage from cities and villages that flows directly into the sea is one of the most important microplastic contamination sources. The seafloor has the largest concentration of microplastics, but it also appears to be a habitat for many fish that people are dependent on for food. Results from many studies have reported that commercial fishing zones were selected as the "hotspots" for microplastic accumulation. Leachates from landfills contain high amounts of plastic waste, and they

are sources of microplastic contamination in groundwater caused by the leaching process. Geogenic contamination sources indicate that the contamination of groundwater is due to geological sources that are responsible for geochemical enriching. According to the study, chemical weathering, evaporation, and weathering of parent minerals (such as arsenopyrite and realgar) are to blame for arsenic, fluoride, manganese, Uranium, sulfur dioxide ($SO_2$), and many trace element contamination. It was determined that As pollution of groundwater had both cancer-causing and non-cancerous health consequences on individuals. Strongly negative loadings were detected for Cu and Mn, and the existence of carbonate rocks is what causes the high levels of hardness. The results of this study suggested that prolonged exposure to polluted groundwater may pose severe health hazards. Studies analyzed several pharmaceutical ingredients concerning their contamination of drinking water and fish. Even though drugs were found in tiny amounts in drinking water, there is no danger to human health from this source of exposure. But, at the same time, it can cause a potential threat to infants 0–3 months old. Significant heavy metals that pollute water naturally or as a consequence of human action include cobalt, cadmium, chromium, copper, mercury, nickel, zinc, and lead. The main heavy metals that presented non-cancer concerns were chromium and lead, but the largest cancer risk was posed by cadmium. According to a study, Pb and Zn are more frequently consumed in food than cutaneous absorption, which is the predominant exposure pathway for cadmium, chromium, copper, nickel, and mercury. Due to substantial amounts of Cd, Ni, and Pb in the soil around some communities or low RfD values, many people are exposed to heavy metal poisoning. Overall, several pollutant causes have a negative impact on human health and livelihood. Although the breadth of the study is expanding, it is unclear what the future holds for health hazards related to water contamination-based research. This study offers a comprehensive overview of the many water contamination sources that can have an impact on the quality of the water, as well as the possible adverse effects on human health. The breadth of interdisciplinary study will be increased by the in-depth material in this review.

**Author Contributions:** Conceptualization, P.B. and S.T.; methodology P.B. and S.T.; software, P.B. and K.D.; resources, P.B. and S.T.; data curation, P.B. and S.T.; writing—original draft preparation, P.B. and S.T.; writing—review and editing, S.T. and K.D.; visualization, S.T. and G.P.; supervision, S.T. All authors have read and agreed to the published version of the manuscript.

**Funding:** This research received no external funding.

**Data Availability Statement:** Not applicable.

**Conflicts of Interest:** The authors declare no conflict of interest.

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
