# Peer review of "Human Health Risks due to Exposure to Water Pollution: A Review"

_water, doi:10.3390/w15142532_

Round 1

Reviewer 1 Report

The article Human health risks due to exposure to groundwater pollution: A global perspective can suitable for publication in WATER.

On the other hand, basing this review article on only 63 publications does not warrant the use of the word global review.

I have several comments to the author. I am sure, that they should have no problems with improving their article before publication.

My main comment- in the introduction, indicate very precisely to whom this review article is addressed

Abstract

The abstract is too general, good for a popular science journal- please include more detailed information

Introduction

Line 44

 "contaminant cocktail”- please avoid spreading such terms

Check line 56 and 57 please

Probably it must be: This review “Human health risks due to exposure to groundwater pollution: A global perspective.”  is a compilation of several…..

This work contains 63 different research paper. What it means contain?? Why only 63?

Figure 1 is of little value. How it was created?

Data collection

Were keywords the only criteria for selecting journals?

Microplastics

The first paper, the second paper- do not use such a term. No more papers about microplastics in the environment?

Conclusions

In my opinion, the chapter should be rewritten. Conclusions are too general.

Even in a review article, conclusions in a serious scientific journal should be more advanced.

“These were reviewed based on the sources of contamination and mode of contamination such as water and air. The associated human health risk assessment was also mentioned in this review paper are-please check and rewrite this sentence

t is a good idea to use native speaker for correction.

Author Response

Dear Editor and Reviewer 

Herewith attached 

Reviewer 2 Report

The authors appeared to have completed a thorough review of the literature regarding human health risks due to exposure to groundwater pollution.  The purpose of the review paper is unclear, though.  A compilation of the information in the literature is not worthy of publication.  The authors should consider the reason for the review and include it in the paper.  Overall, the authors need to provide details within the paper to justify the methods used and results stated.  Specific comments by section are below.

Introduction

In my opinion, authors should elaborate a bit more the introduction section. The introduction lacks of stringency and fails to point out the need for a literature review on the chosen topic. It's unclear what the purposed of this review is.  Are the authors trying to find a niche to study?  What is a "dose-response”?  What was the criteria to select the case studies included in this manuscript as well as the related 63 papers?  How was the perceived study quality addressed?

Other than simply collecting and reviewing literature, and showing the case studies, the more important part of the review is how those 63 selected papers (out of ???? on total) reported their findings and why those papers were more important than those that were eliminated from review. 

L.34. Please avoid expansion of abbr. two times in a row. Only once, when first seen in the main text 

L.56-57. Please erase “This review “Human health risks due to exposure to groundwater pollution: A global perspective.” and start the following sentence as “This review study”

 Methodology

My major concern the overall picture of this manuscript. Who is the intended audience? What is the scope and purpose of this manuscript? Even if the audience, scope and purpose were clear, the "Search strategy" is very weak. The authors should present crucial information concerning the review methodological framework followed (e.g. screening criteria, reasons for period covered, inclusion/exclusion of other material apart from manuscripts etc.). Was any standardized framework used e.g. PRISMA? The authors must refer other papers on the journal as good examples to follow an in-depth presentation of their review framework employed.

L.88. Arsenic is not a heavy metal but metalloid.

A table indicating the different indices (Hazard-, Health-based etc.) reviewed in this study (along with their individual characteristics and use) will be useful to readers.

Conclusion

Overall, the conclusions (as bullet points) of the review are not supported by the evidences provided by the authors in a logical and scientific manner.  The conclusions appear biased.  More scientific evidences are needed to support the conclusions.  Authors should expand the "real" science aspect of literature review.  If the current papers are of "limited usefulness" suggest what can be done to improve future studies.  Without these information, this review will read biased.

Language

Another problem that has distracted my attention was the poor English used. I fully understand that the authors have made serious efforts to present their work in a suitable manner, but the text needs to be thoroughly edited by a native English speaker to improve the flow of thoughts and arguments.

Given that the information presented in the present manuscript are a rich compilation of data mostly accessible by local audience, I believe the overall effort is not valueless but certainly requires major improvements and re-writing.

Author Response

(The authors gave the same response as above.)

Reviewer 3 Report

Overall, the paper explores a human health risks due to exposure to groundwater pollution. The scope and the unique focus is worth the publication, however, it is not acceptable in its current form. Few suggested revisions are provided. In addition, some English editing is needed. 

*To ensure the Editor and Reviewers will be able to recommend that your revised manuscript is accepted, please pay careful attention to each of the comments that have been pasted underneath this email. This way we can avoid future rounds of clarifications and revisions, moving swiftly to a decision.

*Overall, the paper is confusing, it is supposed to be a review paper but the content explores some  perspective or framework. It fails on both fronts. It is recommended to do either a review or a framework and resubmit. It is also recommended to have an English editing.

*For readers to quickly catch your contribution, it would be better to highlight major difficulties and challenges, and your original achievements to overcome them, in a clearer way in abstract and introduction.

*A lot of references are listed. It seems very relevant, but how to do analysis for review or research studies is a big problem usually. 

*There is no method chapter and no clear method description. There should be a clear enough method that enables somebody else to repeat the study.

It is recommended to have an English editing.

Author Response

(The authors gave the same response as above.)

Reviewer 4 Report

Dear Authors
 When I accepted to review your manuscript I expected a critical point of view based on a real overview about contaminants and environmental issues! I am very disapointed by the strategy used (see my comment and questions on your mansucript); blurry sentences and the biais of yourstrategy to provide a clear and well documented review. The consequence is that your manuscript do not do not respect basic and elemental scientific rules. Thus I suggest to the editor to reject it. Hope you good luck in the future. With all my deep respectfull consideration

Author Response

(The authors gave the same response as above.)

Round 2

Reviewer 1 Report

Article was improved. Still my biggest comment is is related to the amount of literature used in the article.

Author Response

Point 1: Article was improved. Still my biggest comment is is related to the amount of literature used in the article.

Response 1: Thank you for your value able comment. In this review paper, the literature used contains those topics which purely deal with health risk calculations based on different formulae.

Reviewer 2 Report

After thoroughly reviewing the revised manuscript, I believe that the authors made only a few efforts to revise it according to previous comments. In this context, many of the flaws raised in the previous rounds of revision are still valid e.g. still As is considered as a heavy metal in the review study (L.359),  nothing is mentioned regarding the quality addressed for the final 63 papers investigated, L.56-57 and others.

moderate revisions

Author Response

RESPONSE TO REVIEWER 2 COMMENT (BLUE)

Point 1: In this context, many of the flaws raised in the previous rounds of revision are still valid e.g. still As is considered as a heavy metal in the review study (L.359).

Response 1: Thank you for your suggestion. I have corrected all the places as As (Arsenic) as a metalloid and corrected this review paper also.

Point 2: nothing is mentioned regarding the quality addressed for the final 63 papers investigated

Response 2: The main criteria used to select the case studies were health risk assessment in connection with various types of water contamination. Those 63 selected papers were more important because they had human health risk evaluation for the water contamination-affected area.

Point 3: L.56-57 and others.

Response 3: As per the reviewer’s suggestion line 56-57 has been rectified.

Reviewer 3 Report

I think this manuscript is ready to publish

I think minor editing of English language required

Author Response

RESPONSE TO REVIEWER 3 COMMENT (GREEN)

Point: I think minor editing of the English language required

Response: The authors have corrected the minor English language mistake in the revised manuscript.

Reviewer 4 Report

Dear authors

You tried to ameliorate your manuscript which from my narrow point of view of scientific is not enough. This study is at the level of an environmental science report based (around 80%) on articles focused essentially on Asia and the Middle East, and cannot reflect the degree of contamination of water resources (I don't take in consideration the atmospheric contaminant which is very poor) or the possible solutions (where they exist) to remedy the situation. The conclusion is quite astonishing in several categories. Air quality is analysed on 3 targets without taking into account PM<10, NOx, VOC dioxine from waste treatment plant based on incineration process, for example). Pharmaceuticals (which ones exactly?) and concluding that there is no danger to human health is absolutely false! How can you write this!!!! What about oestrogens, antibiotic resistance and the fact that the routes of contamination are both water and air? As for plastics, they are limited to fish and fishing grounds and ingestion. What about the compounds present in plastics (trace metals, organic and inorganic additives, and the recent studies on nanoplastics crossing supposed biological barriers? There's nothing on emerging substances, in particular PFOAs, even though recent studies (https://doi.org/10.1080/10408444.2021.1888073). Could you at least change the title to reflect the limitations of your review and try to offer an honest and accurate conclusion? Of course air, pharmaceutical and plastic chapter have to be revised by taking into account more govermental and NGO reports, published papers their conclusions. My opinion about your manuscript has not changed basically even if after major revisions it could be published and I leave it to the editor to assume his editorial responsibility.
Best regards

Author Response

Point 1: This study is at the level of an environmental science report based (around 80%) on articles focused essentially on Asia and the Middle East, and cannot reflect the degree of contamination of water resources (I don't take in consideration the atmospheric contaminant which is very poor) or the possible solutions (where they exist) to remedy the situation.

Response 1: Respected reviewer in this review article the authors have not mentioned anything regarding Air.

Point 2: The conclusion is quite astonishing in several categories. Air quality is analysed on 3 targets without taking into account PM<10, NOx, VOC dioxine from waste treatment plant based on incineration process, for example).

Response 2: Respected Reviewer thank you for suggestions, in this review article air is not included. This review paper only discusses water contamination.

Point 3: Pharmaceuticals (which ones exactly?) and concluding that there is no danger to human health is absolutely false! How can you write this!!!! What about oestrogens, antibiotic resistance and the fact that the routes of contamination are both water and air?

Response 3: Respected sir there are several published articles that mention the pharmaceutical hazard but the authors have taken those articles which give a qualitative view about human health hazard through formulae considering a person’s amount of intake of water. The following articles mention several pharmaceuticals and they conclude that those pharmaceuticals are not posing threats.

  • Cunningham, V. L., Binks, S. P., & Olson, M. J. (2009). Human health risk assessment from the presence of human pharmaceuticals in the aquatic environment. Regulatory toxicology and pharmacology53(1), 39-45.
  • [47] J. Houtman, J. Kroesbergen, K. Lekkerkerker-Teunissen, and J. P. van der Hoek, “Human health risk assessment of the mixture of pharmaceuticals in Dutch drinking water and its sources based on frequent monitoring data,” Science of the Total Environment, vol. 496, pp. 54–62, Oct. 2014, doi: 10.1016/j.scitotenv.2014.07.022.
  • [48] W. Schwab et al., “Human pharmaceuticals in US surface waters: A human health risk assessment,” Regulatory Toxicology and Pharmacology, vol. 42, no. 3, pp. 296–312, Aug. 2005, doi: 10.1016/j.yrtph.2005.05.005.
  • [49] de Jesus Gaffney, C. M. M. Almeida, A. Rodrigues, E. Ferreira, M. J. Benoliel, and V. V. Cardoso, “Occurrence of pharmaceuticals in a water supply system and related human health risk assessment,” Water Res, vol. 72, pp. 199–208, Apr. 2015, doi: 10.1016/j.watres.2014.10.027.

Point 4: As for plastics, they are limited to fish and fishing grounds and ingestion. What about the compounds present in plastics (trace metals, organic and inorganic additives, and the recent studies on nanoplastics crossing supposed biological barriers? There's nothing on emerging substances, in particular PFOAs, even though recent studies (https://doi.org/10.1080/10408444.2021.1888073).

Response 4: Respected reviewer There are several published articles that highlight microplastics, their types, and the risk it poses to the environment. But in this review article, the authors have focused only on those kinds of literature which signify human health risk in a quantitative manner using human health risk formulae. Hence the authors got few works of literature only on the concerned topic of “Microplastic”

Point 5: Could you at least change the title to reflect the limitations of your review and try to offer an honest and accurate conclusion?

Response 5: As per Reviewer’s suggestion the title of the review has been changed to “Quantitative Analysis of Human health risks due to Exposure to water pollution”

Point 6: Of course air, pharmaceutical and plastic chapter have to be revised by taking into account more govermental and NGO reports, published papers their conclusions.

Response 6: Respected reviewer the air chapter is not included in the review article. The Pharmaceuticals and Microplastic chapter has been revised, but no government or NGO reports give a quantitative report on human health due to exposure to water contamination. Hence those articles are included which gives a quantitative value to health risk.
